# The Effect of Just-World Beliefs on Cyberaggression: A Moderated Mediation Model

**DOI:** 10.3390/bs13060500

**Published:** 2023-06-14

**Authors:** Qingsong Sang, Qi Kang, Kun Zhang, Shouli Shu, Lijuan Quan

**Affiliations:** 1School of Educational Science, Anhui Normal University, Wuhu 241000, China; kangqi@hnuu.edu.cn (Q.K.);; 2School of Information Engineering, Huainan Union University, Huainan 232000, China

**Keywords:** just-world beliefs, self-control, cyberaggression, gender

## Abstract

(1) Background: To examine the relationship among just-world beliefs, self-control, and cyberaggression among college students. (2) Methods: A total of 1133 college students were surveyed using the just-world belief scale, self-control scale, and cyberaggression scale. (3) Results: The results showed that college students with low levels of belief in justice frequently showed cyberaggression; belief in a just world directly and negatively predicted cyberaggression, and indirectly predicted student cyberaggression through self-control; gender moderated the indirect effect of self-control on cyberaggression and the direct effect of belief in a just world on cyberaggression. (4) Conclusions: Belief in a just world significantly and negatively predicts cyberaggression; self-control has an indirect significant effect on cyberaggression; the direct effect of belief in a just world on cyberaggression and the mediating effect of self-control on this association are moderated by gender.

## 1. Introduction

The creation of the Internet is of major importance in the development of human civilization. The term “cyberaggression” refers to hostile behavior caused by inhibitive characteristics, and involves the use of networks or mobile electronic devices to write obscene or insulting communications that provoke, threaten, and harm a person or group [1]. Unlike the direct use of objects to cause physical injury, cyberaggression involves threatening and abusing others through obscure texts or negative pictures/symbols; this leads to psychological pain and long-term negative effects [2]. College students are common users of network platforms, and cyberaggression occurs frequently in college students. It has been found that the incidence of network attacks among college students is about 60% [3], and about 55% of college students have suffered network attacks [4]. Cyberaggression can cause many physiological and psychological problems, such as anger, depression, anxiety, eating disorders, alcohol addiction, and even suicidal tendencies [5,6,7]. However, compared to traditional forms of aggression, cyberaggression has been less studied [8]. Therefore, it is necessary to further explore the psychological mechanisms underlying cyberaggression. In addition to helping to provide a psychological foundation to guide network supervision and network behavior, this would help to improve network-related moral reasoning in college students, improve the moral environment of network use, and develop a better online public mental health environment.

Individuals who believe in a just world firmly believe that the world they live in is just, and that their own and others’ gains and losses are determined and distributed by the world. Belief in a just world can provide individuals with a sense of security and control, and can moderate individual cognition, behavior, and emotion [9]. Generally, individuals with high levels of belief in justice are more inclined to believe that the world they live in is just, always have high expectations of life, are able to maintain a positive attitude about the future, voluntarily struggle for an ideal life, complete goals in a way that conforms to social norms, and strive to avoid all problematic behaviors that hinder the realization of their goals [6,10,11,12]. However, real life is often characterized by injustice. Serious threats to an individual’s belief in self-justice reduce their sense of control over life and the surrounding environment and cause anxiety, anger, and hostility, thus stimulating aggression [13,14,15]. Moreover, individuals with low levels of belief in justice may use irrational cognitive strategies to reconstruct a sense of justice at the cognitive level. That is, such individuals tend to re-evaluate and interpret a victim’s character, morality, and other aspects to try to determine the cause of their unjust treatment. During reconstruction of justices, a hostile attitude toward others and attribution bias activate individuals’ implicit aggression schemas. In the network environment, this implicit aggression schema becomes more and more intense because of the anonymity and freedom of network communication, which induces more cyberaggression [16,17].

Self-control is the ability to restrain impulsive thoughts, regulate negative emotions, and regulate personal behaviors [18]. Research indicates that people with low self-control are more likely to show problem behaviors, such as crime, drug abuse, and alcoholism; in contrast, people with high self-control have healthy eating habits, good interpersonal relationships, and fewer diseases [19,20,21,22]. Denson et al. [23] identified self-control failure as an important predictor of aggression; experiments have shown that in a provocative situation (noise interference caused by others), individuals who receive self-regulation training feel less angry and retaliate against others less. The dual-process model of self-control posits that when faced with temptation, individuals are prompted to make an emotional response to satisfy their impulsive desires; at the same time, they are stimulated to consciously evaluate and control the consequences of their impulses. However, if the intensity of the impulse motivation exceeds the cognitive effort to resist or regulate the impulse, the individual’s attempts at self-control fail; this causes a series of problem behaviors [24]. For example, an individual with a fluctuating or unstable belief in justice will often use the cognitive model of “everything that happens is reasonable” to rationalize the unjust events that happen to victims. In addition, they will often use slander, ridicule, satire, and other methods to argue that others deserve their punishment, to maintain long-term stability of their belief in justice. Over time, this type of hostility to others uses more cognitive resources. When it exceeds the limit of self-regulation, it makes individuals less sympathetic. Moreover, in the network environment of free speech and anonymity, individuals may express their impulses as cyberaggression to vent their emotions [3,16,25].

Some studies have also identified differences in cognitively ambiguous cues between male students and female students. Compared with women, men who are treated unjustly are more motivated to seek revenge when using the Internet and show a more obvious tendency to attack [26]. This could be explained by evolutionary differences. Generally, men are expected to be more competitive and in control, whereas women are expected to be more obedient to norms, honest, and kind. In interpersonal communication, women are more likely to care for and understand others, are more willing to cooperate with others, and show more altruistic will; in contrast, men are more concerned about personal interests, tend to compete with others, and are more likely to use irrational cognitive strategies to maintain their belief in justice when experiencing cognitive distress over the treatment of others. Women often use self-regulation to control the development of risk-taking behavior, and transform social control into self-control, thus reducing the likelihood of crime; thus, men show more aggression [27].

To summarize, the belief in a just world theory, the dual-process model of self-control, and the GAM were used to generate the following three hypotheses, which were tested in this study: H1—the level of college student cyberaggression will be positively affected by a high level of belief in justice; H2—there will be an obvious indirect effect (mediating effect) of self-control; H3—the mediating effect of self-control will be moderated by gender. A hypothetical model was generated and is shown in Figure 1.

## 2. Methodology

### 2.1. Subjects

Cluster sampling was used to survey students from four universities in Anhui and Henan provinces, China. A total of 1308 questionnaires were distributed to various classes. After excluding blank and invalid questionnaires and missing data, 1133 valid questionnaires (86.62%) were counted. Subjects who returned valid questionnaires comprised 460 male students (40.6%), 673 female students (59.4%), 344 freshmen (30.4%), 217 sophomores (19.2%), 318 juniors (28.1%), and 254 seniors (22.4%). The age ranged from 17 to 25 years, with a mean was 20.67, and the standard deviation was 1.59 (average was 20.67 ± 1.59).

### 2.2. Research Tool

#### 2.2.1. The Just-World Belief Scale

The just-world belief scale developed by Dalbert (1999) was used. The scale consists of two subscales: the general just-world belief scale (GBJW) and the personal belief in a just world scale (SBJW). The general just-world belief scale measures people’s judgments about the fairness of what happens to others, and the personal belief in a just world scale measures people’s judgments about the fairness of what happens to them.

The scale comprises a total of 13 items. The personal belief in a just world scale’s belief in self-justice subscale consists of seven questions, for example, “I believe most things that happen in my life are fair”. The general just-world belief scale comprises six questions, for example, “Generally others are treated fairly”. Responses were scored on a 6-point Likert scale, where 1 indicates complete disagreement and 6 indicates complete agreement, on a scale from 1 to 6; the average score of each item of the scale was obtained, with higher scores indicating higher levels of belief in justice. The α coefficient of the scale was 0.90, and the α coefficients of the two subscales were 0.84 and 0.87, respectively. McDonald’s omega coefficient was 0.90.

#### 2.2.2. Self-Control Scale

The SCS-19 College Student Self-Control Scale developed by Tangney et al. (2004) and revised by Guo Yongyu et al. was used. The scale comprises 19 questions in 5 dimensions, namely resisting temptation, healthy habits, limiting entertainment, impulse control and focus on work, of which 4 (questions 1, 5, 11 and 14) were forward-scoring questions, while the remaining 15 items were scored in the reverse direction. Responses were scored on a 5-point Likert scale; the item options were “completely inconsistent, somewhat inconsistent, uncertain, fairly consistent, and completely consistent,” rated 1, 2, 3, 4, and 5, respectively; higher scores indicate greater self-control. The α coefficient of the scale in this study was 0.85, and McDonald’s omega coefficient was 0.84.

#### 2.2.3. Cyberaggression Scale

The Scale for Adolescent Internet Deviance (sAID) compiled in Li Dongmei’s (2008) doctoral dissertation was adopted; this scale contains 3 basic dimensions with a total of 35 items: 20 items of aggressive online behavior, 9 items of online pornographic behavior, and 5 items of online cheating. Among the three dimensions, online aggressive behavior is subdivided into four more dimensions, namely 6 items of aggression, 5 items of irritability, 5 items of hostility, and 4 items of conflict. In this study, only one dimension, namely aggression, with 6 questions in total, was selected, for example, “On the Internet, I will say something on purpose to make others sad”. Responses were scored on a 5-point Likert scale, where ratings range from 1 (never) to 5 (always); higher scores indicate more frequent cyberaggression. The α coefficient of the scale in this study was 0.87. McDonald’s omega coefficient was 0.87.

### 2.3. Program and Data Processing

Sequential group measurement was used. Before subjects filled in the questionnaire, the examiner explained the study aim and emphasized that responses would be anonymous and that the scores are neither good nor bad. The subjects were asked to complete the questionnaires independently and to provide their honest responses. After the data collection was complete, SPSS19.0 software and the PROCESS macro version 3.3 compiled by Hayes (2018) were used for statistical analysis, and the moderated mediation effect was tested. In this study, the score for each variable was calculated from the standard score of the mean score for each question of the scale.

## 3. Results and Analysis

### 3.1. Common Method Bias Test

Exploratory factor analysis of all variables was conducted following Harman [28] and using the single factor test. The test identified seven factors with a latent root greater than 1. Of these, the first factor explained 20.34% of the variance, less than the critical value of 40% (i.e., there was no serious method bias).

### 3.2. Correlations among Variables

The correlation analysis (see Table 1) showed that belief in a just world significantly and negatively correlated with cyberaggression (r = −0.24, *p* < 0.01), belief in a just world significantly and positively correlated with self-control (r = 0.21, *p* < 0.01), and cyberaggression significantly and negatively correlated with self-control (r = −0.30, *p* < 0.01).

### 3.3. Moderated Mediation Effect Test

After controlling additional variables such as age and grade, a simple mediation effect analysis was carried out using Model 4 in the PROCESS macro 3.3 version compiled by Hayes (2018) [29]. As shown in Table 2, belief in justice had a significant negative effect on cyberaggression (*B* = −0.22, *t* = −8.08, *p* < 0.01), and this effect was retained (*B* = −0.17, *t* = −6.28, *p* < 0.01) when the mediation variables were included. Belief in justice significantly and positively predicted self-control (*B* = 0.20, *t* = 6.93, *p* < 0.01), and cyberaggression significantly and negatively predicted self-control (*B* = −0.26, *t* = −9.65, *p* < 0.01). In addition, the bootstrap 95% confidence interval of the mediating effect of self-control did not include 0 (see Table 3), which indicates that self-control mediated the association between belief in justice and cyberaggression. The direct effect (−0.170) and mediating effect (−0.053) accounted for 76.23% and 23.77% of the total effect (−0.223), respectively.

The PROCESS macro Model 15 compiled by Hayes (2018) [30] was used to test the moderated mediation effect. The results are shown in Table 4 and Table 5. When gender was added to the model, which moderated the direct effect of belief in a just world on cyberaggression and the indirect effect of self-control on cyberaggression, (*B* = 0.13, *t* = 2.48, *p* < 0.05; *B* = 0.14, *t* = 2.60, *p* < 0.01), indicating that gender moderated not only the direct effect of belief in a just world on cyberaggression, but also the indirect effect of self-control on cyberaggression. To clarify the nature of the interaction terms, we used a simple slope test to analyze the moderating effect of gender (see Figure 2 and Figure 3).

Generally, male students were more likely than female students (*M* = 10.53 > 8.16, *t* = 11.16, *p* < 0.01) to exhibit cyberaggression. According to Figure 2, the negative predictive effect of belief in a just world on cyberaggression is higher for male students (*B* = −0.25, *t* = −6.10, *p* < 0.001) than for female students (*B* = −0.11, *t* = −3.04, *p* < 0.01) in the context of the interaction effect between belief in a just world and gender. Similarly, as depicted in Figure 3, the negative predictive effect of self-control ability on cyberaggression is more significant for male students (*B* = −0.34, *t* = −8.49, *p* < 0.001) than for female students (*B* = −0.20, *t* = −5.47, *p* < 0.001) in the context of the interaction effect of self-control ability and gender.

## 4. Discussion

Based on previous study findings, the theory of belief in a just world, and the dual-process model, a moderated mediation model was constructed, with self-control as the mediating variable and gender as the moderating variable. We aimed to clarify how belief in a just world affected college student cyberaggression (the mediating effect of self-control), as well as examining which factors affected the significance of the effect of belief in a just world on cyberaggression (the moderating effect of gender). The results have theoretical and practical implications for further research on the relationship between cyberaggression and psychological and personality differences, as well as for efforts to help college students to use the Internet sensibly and psychologically adapt to virtual communication.

### 4.1. The Mediating Effect of Self-Control

After establishing that belief in a just world negatively predicts college student cyberaggression, we further tested the mediating effect of self-control on the association between belief in a just world and cyberaggression in college students. The results showed that students with low levels of belief in justice frequently initiated network aggression owing to a lack of self-control, which is consistent with the results of previous studies [25,31,32]. Dalbert et al. [33] suggested that belief in a just world can help people to better adapt to the environment and to view the world in a positive and meaningful way; this enables them to more confidently determine their value and direction in life. Although individuals may experience tension and anxiety in response to stressful life events (e.g., unjust treatment), if they take the initiative to use cognitive control resources to re-evaluate the situation, this will interrupt the process of hostile reflection on negative stimulation and suppress the anger, tension, and other negative emotions generated by hostile reflection, thus preventing aggression [34]. In addition, individuals with high levels of belief in justice have strong motivation to achieve long-term goals. They are also better at delaying gratification, even in dangerous situations, consciously avoiding the negative interference of victim affect on rational attribution and reducing the risk of impulsive aggression. In contrast, individuals with a weak belief in justice increasingly become unjust, and generally believe that regardless of whether they work hard or not, they will not necessarily achieve their future goals. Therefore, they are less willing to adhere to and achieve long-term goals and are often limited to immediate interests; they may even engage in irrational aggression to achieve immediate gratification [12,35]. Particularly in online social interaction, it is easy for such individuals to unduly categorize fuzzy cues (i.e., ambiguous information) as threatening, and thus reflect and interpret these stimuli in a hostile way, resulting in attentional bias, cognitive bias, and cyberaggression owing to excessive self-depletion [34,36,37]. This explanation is consistent with the generalized tension theory [38]. Because individuals with low levels of belief in a just world have a low sense of control over life and trust in others, anonymous online social interaction increases the fuzziness of communicative information in the absence of physical cues, which leads to anxiety and tension. Such individuals have a very strong sense of self-protection. Encountering verbal provocation from others automatically activates aggressive thoughts and negative thinking, resulting in the narrowing of attentional scope and weakening of inhibitory function. Aggression (even violent behavior and other extreme responses) is often used to restore a sense of balance and alleviate the negative effects of tension; this helps the person to return to a normal bodily balance. Individuals with low levels of belief in justice who have negative experiences are particularly likely to have a strong desire to obtain positive attention and fair treatment when using virtual networks. If their desire is blocked, they are more likely to be disturbed by negative stimulation and to express hostile impulses. The anonymity and openness of the network environment and the lack of social cues (e.g., bodily cues, facial expressions) reduces individuals’ awareness of impulsive behavior, making them less likely to exercise self-restraint and self-control; this leads to more intense aggression [3,13,35].

### 4.2. The Moderating Effect of Gender

Based on the GAM theory, a moderated mediation model was constructed to investigate the moderating effect of gender on the relationship between belief in a just world and self-control and cyberaggression. The results showed that the relationship between belief in a just world and cyberaggression was moderated by gender. In addition, gender moderated the second half of the mediating effect; that is, the relationship between self-control and cyberaggression.

Specifically, belief in justice more significantly affected cyberaggression in male college students than in female college students. The results showed that male and female students differed in the cognitive mechanism (belief in a just world) of cyberaggression, which is consistent with previous study findings [39]. First, compared with men, women are considered to be better at expressing their emotions, more likely to self-moderate negative emotions induced by negative life events, and less likely to engage in deviant behaviors [40]. Second, the goal consistency theory suggests that women have greater behavioral control and goal behavior motivation than men [30]. In particular, women with high levels of belief in justice have a strong desire to achieve long-term goals and yearn for a better future, and they are more willing to improve their self-worth in a fair society through their own efforts. Even when they encounter unjust treatment, they can consciously use rational strategies to re-evaluate the situation, make reasonable attributions, and reduce aggression caused by the victim effect. The GAM also posits that gender, as a stable personal factor, is an important cause of cyberaggression [31]. The concept of cynicism, which is often linked to a competitive outlook on life, tends to be deeply entrenched in men who harbor low levels of belief in justice. When they are in cyberspace, which is characterized by freedom of speech, they are more likely than women to use irrational cognitive strategies toward others to obtain a sense of fairness and control. For example, they may blame and belittle others to rationalize the causes of injustice; this generates more cyberaggression over time. Some studies show that compared with women, the more injustice that men encounter in real life, the more likely they are to take revenge when using the Internet [26]. In addition, according to social cognitive processing theory, the network environment allows individuals who encounter negative events to regain a sense of justice. However, men with low levels of belief In justice feel inferior, owing to a lack of trust in others and a sense of control over real life. The Internet provides such individuals with a comfortable environment that provides emotional sustenance. Although these individuals are eager to restore fairness to the network environment, during network communication, they overinterpret vague information or expressions as threatening. When recalling communicative exchanges, they experience strong aversion reactions and hostile reflection on others’ behavioral motivations, which leads to cyberaggression [34].

The present findings also showed that male students with low self-control tend to initiate more obvious cyberaggression. The results showed that gender, as a personality trait, has a moderating effect on the influence of other variables on cyberaggression. Previous studies have shown that gender, as a stable personality trait, has a potential effect on individual behavior and psychological adaptation [41]. In the process of socialization, women are usually expected to show warmth, solicitude, and care for others. When women express aggression inconsistent with their social roles, they experience more guilt and shame [38]. In contrast, men attach less importance to social relations and are subject to more active and competitive stereotypes. However, such expectations may lead them to be more inclined to violate social norms in pursuit of personal interests and show more problematic behaviors [42]. Men are also considered to show less emotional expression than women. In particular, when men experience negative emotions (e.g., sadness, pain), they are more likely to repress them [40]. However, interestingly, when men participate in abnormal behaviors, such as shoplifting and attacking others, they experience more positive emotions than women. When they recall violations that they have committed, they are more tolerant and confident than women [42]. Compared with male college students, female students with high self-control tend to fully assess the risk of Internet cyberaggression and consider moral constraints and social customs, so do not tend to act impulsively; compared with female students, male students with low self-control make insufficient use of cognitive resources, experience more self-depletion, show more cognitive impulsivity [43], think less deeply about moral offences, and often express impulsive behaviors owing to a failure of self-control [24] when using the Internet.

### 4.3. Educational Significance

This study used questionnaire responses to assess the psychological mechanisms underlying the generation of cyberaggression in college students. The construction of a moderated mediation model could greatly inform strategies to help college students use the Internet in a more rational and balanced way, and thus improve their mental health.

First, belief in a just world is a protective factor that maintains personal physical and mental health, and prevents individuals from misunderstanding and misinterpreting unjust treatment and committing violations. Belief in a just world not only provides students with a positive perspective on the world, but also intensifies their love for and confidence in life. Adolescence is a critical period for the formation of belief in a just world. During this period, teachers and parents should focus on improving students’ perceptions of a greater sense of justice, prompting them to form a positive attitude toward system power and encouraging them to voluntarily comply with laws and regulations. Additionally, a reward and punishment mechanism to maintain justice should be established to mete out severe punishment to offenders who violate the principles of justice. Second, gratitude education should be promoted to improve the sense of fairness perceived in interpersonal interactions, enhance friendship and trust between classmates, and avoid individual blaming and belittling of others to defend self-righteous beliefs; this would help to reduce aggression. In addition, it is necessary to continuously improve students’ awareness of self-control, encourage them to use rational cognitive strategies to positively interpret and evaluate stressful situations, and consciously use cognitive control resources to reduce impulsive motivation, thereby reducing the damage caused by aggression to others. Finally, colleges should provide mental health education to students and parents, focus on cultivating college students’ interpersonal skills, and encourage students to seek psychological assistance from professionals. For students who feel distressed and anxious owing to negative life events, psychological therapies such as group counseling or cognitive correction training could be used to help them correctly recognize and actively respond to stressful events. This would reduce aggression caused by misinterpretation and hostile reflection regarding stressful events.

### 4.4. Reflection and Prospects

With regard to clinical implications, cyberaggression response is a systematic project, which requires the joint efforts and all-round cooperation of the government, schools, and families. The government should establish and improve the network laws and regulations to combat cyberaggression; schools should carry out good work in technology monitoring, filtering aggression information, and providing timely psychological counseling for students who experience cyberaggression; parents should manage their minor children’s mobile phone use and educate them to use the Internet properly.

There were some study limitations. First, a questionnaire survey was used to collect data; such methods differ from experimental methods and are less rigorous. In order to ensure the objectivity and accuracy of the research results, the follow-up research can consider combining behavioral experiments and ERP experiments. A cross-sectional design was used and data from a large-scale survey project were analyzed. However, the selected sample was limited (college students). Therefore, we cannot make causal inferences about the relationship between the variables. To confirm the present findings, additional studies are needed that use tracking designs. Thirdly, due to the limitations of practical conditions, the objects of this study only involve college students from four universities in Anhui Province, so the samples cannot reflect the overall characteristics well. Later research can expand the region of research object selection and increase the representativeness of the research samples. Fourthly, participation in cyberaggression may have been underreported because of the tendency of individuals to provide socially desirable answers. In addition, recall bias may have happened, which may also have caused a certain bias in the study results. Fifthly, this study only discusses the influence of gender and self-control on the relationship between belief in a just world and cyberaggression behavior, but it is only a preliminary exploration. Many questions still need to be solved, such as the mechanism of self-control and further intervention research.

## 5. Conclusions

After controlling for additional variables such as age and grade, this study found that belief in a just world can significantly negatively predict college students’ cyberaggression behavior, and self-control plays a mediating role between belief in a just world and cyberaggression. Gender serves as a moderating factor in two key relationships: firstly, it impacts the direct predictive effect of belief in a just world on cyberaggression; secondly, it also moderates the mediating effect of self-control on the link between belief in a just world and cyberaggression.

## Figures and Tables

**Figure 1 behavsci-13-00500-f001:**
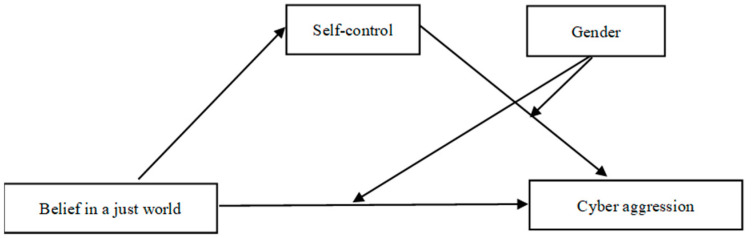
Hypothesis model.

**Figure 2 behavsci-13-00500-f002:**
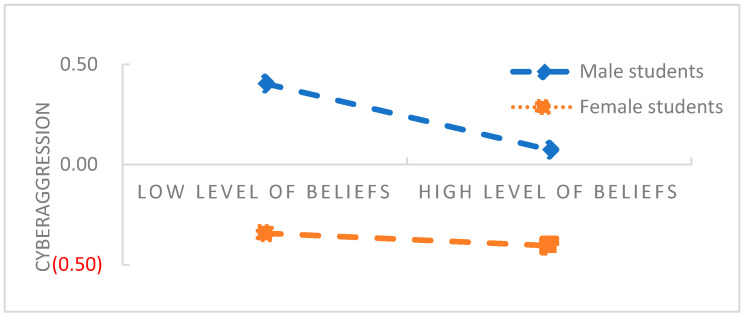
Moderation of direct effect by gender (standardized).

**Figure 3 behavsci-13-00500-f003:**
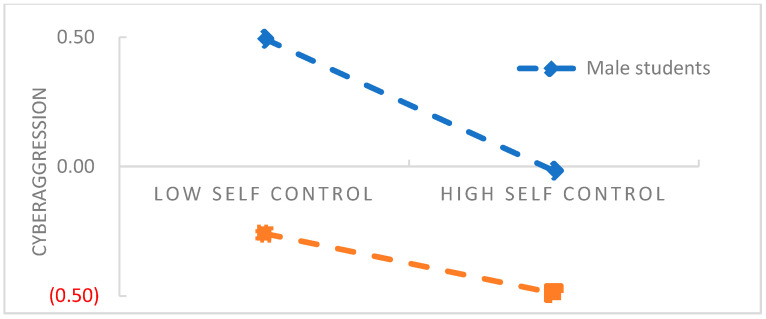
Moderation of indirect effect by gender (standardized). (

 Female students).

**Table 1 behavsci-13-00500-t001:** Descriptive statistics and correlation analysis results (N = 1133).

	*M*	*SD*	Gender	Age	Belief in a Just World	Self-Control	Cyberaggression
Gender	1.59	0.49	1				
Age	20.67	1.59	−0.14 **	1			
Belief in a just world	52.10	10.40	0.05	0.05	1		
Self-control	61.57	9.46	0.02	0.07 *	0.21 **	1	
Cyberaggression	9.12	3.69	−0.31 **	0.00	−0.24 **	−0.30 **	1

Note: * *p* < 0.05; ** *p* < 0.01.

**Table 2 behavsci-13-00500-t002:** Test of mediating effect of self-control.

Regression Equation		Fit Index			Significance of Correlation Coefficient	
Result Variable	Predictor Variable	*R*	*R* ^2^	*F*	*B*	*t*
Cyberaggression		0.39	0.15	49.54 **		
	Gender				−0.31	−11.01 **
	Age				−0.04	−0.84
	Grade				0.01	0.19
	Belief in a just world				−0.22	−8.08 **
Self-control		0.21	0.05	13.53 **		
	Gender				0.03	0.51
	Age				0.04	1.48
	Grade				−0.01	0.30
	Belief in a just world				0.20	6.93 **
Cyberaggression		0.46	0.21	61.49 **		
	Gender				−0.62	−11.30 **
	Age				−0.01	−0.45
	Grade				0.01	0.10
	Belief in a just world				−0.17	−6.28 **
	Self-control				−0.26	−9.65 **

Note: ** *p* < 0.01.

**Table 3 behavsci-13-00500-t003:** Analysis of all effects.

	Effect Size	Boot SE	Bootci Lower Limit	BootCI Upper Limit	Relative Effect Value
Total effect	−0.233	0.028	−0.277	−0.169	
Direct effect	−0.170	0.027	−0.233	−0.117	76.23%
Mediating effect of self-control	−0.053	0.009	−0.071	−0.036	23.77%

**Table 4 behavsci-13-00500-t004:** Moderated mediation effect test.

Regression Equation (N = 1133)		Fit Index			Significance of Correlation Coefficient	
Result Variable	Predictor Variable	*R*	*R* ^2^	*F*	*B*	*t*
Cyberaggression		0.48	0.23	46.80 **		
	Gender				−0.61	−11.31 **
	Age				−0.01	−0.49
	Grade				0.01	0.27
	Belief in a just world				−0.17	−6.11 **
	Self-control				−0.26	−9.46 **
	Belief in a just world × gender				0.13	2.48 *
	Self-control × gender				0.14	2.60 **

Note: * *p* < 0.05; ** *p* < 0.01.

**Table 5 behavsci-13-00500-t005:** Comparison of gender differences.

Gender	Mediating Effect Value	Boot SE	BootCI Lower Limit	BootCI Upper Limit
Male students	−0.069	0.013	−0.096	−0.045
Female students	−0.040	0.008	−0.058	−0.026
Difference	0.028	0.012	0.006	0.053

## Data Availability

The raw data supporting the conclusions of this article will be made available by the authors, without undue reservation.

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
