# Peer review of "The Effect of Just-World Beliefs on Cyberaggression: A Moderated Mediation Model"

_behavsci, 2023, doi:10.3390/bs13060500_

Round 1
Reviewer 1 Report
Dear authors: First of all, congratulations for your research on Just-world beliefs, self-control and cyber-aggression, I find it quite interesting how you have proposed it as a mediation model with the self-control variable, with both the direct and indirect effect being moderated with gender. .
However, I see the need to improve the manuscript because I see gaps in it, such as the following:
1. I have seen that they do not mention the existence of missing in the scales used, only that those subjects or respondents who have carried out the survey have been included. I would like to mention if there have been these missing data and their treatment.
2. I see the need to include some more internal consistency and validity indices such as Mcdonald's omega or composite reliability, as well as the extracted mean variance. I have also been able to verify how Cronbach's alpha is some dimensions of the Tangney et al Self-control scale, they are not high, rather acceptable, even poor.
3. I think it would be appropriate to mention or comment on how the scores for each of the scales or variables have been calculated.
Author Response
- I have seen that they do not mention the existence of missing in the scales used, only that those subjects or respondents who have carried out the survey have been included. I would like to mention if there have been these missing data and their treatment.
The author response: Thank you for your suggestion. Following your recommendations, we filtered and removed three missing data. Please see the Method section of our revised manuscript.
2.1 Subjects
Cluster sampling was used to survey students from four universities in Anhui and Henan provinces, China. A total of 1,308 questionnaires were distributed to various classes. After excluding blank and invalid questionnaires and missing data, 1,133 valid questionnaires (86.62%) were counted. Subjects who returned valid questionnaires comprised 460 male students (40.6%), 673 female students (59.4%), 344 freshmen (30.4%), 217 sophomores (19.2%), 318 juniors (28.1%), and 254 seniors (22.4%). The age ranged from 17 to 25 years which the mean was 20.67 and the standard deviation was 1.59 (average was 20.67±1.59) .
- I see the need to include some more internal consistency and validity indices such as Mcdonald's omega or composite reliability, as well as the extracted mean variance. I have also been able to verify how Cronbach's alpha is some dimensions of the Tangney et al Self-control scale, they are not high, rather acceptable, even poor.
The author response: Thank you for your suggestion. We recalculated the α and McDonald's omega coefficients based on the deleted data, at 0.85 and 0.84, respectively.
2.2.2 Self-Control Scale
The SCS-19 College Student Self-Control Scale developed by Tangney et al. (2004) and revised by Guo Yongyu et al. was used. The scale comprises 19 questions, of which 4 are forward-scoring questions. Responses are scored on a 5-point Likert scale; higher scores indicate greater self-control. The α coefficient of the scale in this study was 0.85, and the McDonald's omega coefficients is 0.84.
- I think it would be appropriate to mention or comment on how the scores for each of the scales or variables have been calculated.
The author response: Thank you for your suggestion. Following your suggestion, we have added how to calculate the score for each variable.
2.3 Program and data processing
Sequential group measurement was used. Before subjects filled in the questionnaire, the examiner explained the study aim and emphasized that responses would be anonymous and that the scores are neither good or bad. The subjects were asked to complete the questionnaires independently and to provide their honest responses. After the data collection was complete, SPSS19.0 software and the PROCESS macro version 3.3 compiled by Hayes (2018) were used for statistical analysis, and the moderated mediation effect was tested. In this study, the score for each variable was calculated from the standard score of the mean score for each question of the scale.
Reviewer 2 Report
Thank you for inviting me to review this interesting manuscript. I have some comments outlined below:
Introduction
1. some statements in the introduction need supportive citations
2. please highlight the knowledge gap.
3. Can you shorten the introduction?
Methods
1. Why did you select this type of sampling?
2. How did you calculate the sample size?
3. Where are the inclusion and exclusion criteria?
4. elaborate more on the measurement tools. What is the total score of each tool and what does it indicate?
5. the procedure of data collection needs more details. e.g., how did you approach the target population and recruit your sample?
6. Where are the ethical considerations of your study? e.g., IRB approval and informed consent procedures?
discussion
State the clinical implications and recommendations for future research
highlight the strengths of your study before the limitations and think of more limitations
conclusion
revise your conclusion to be stated in a narrative way
Author Response
Review Report 2
Introduction
- some statements in the introduction need supportive citations
Author response: Thank you for your suggestions. According to your suggestions, we have added some supportive citations in the introduction.
- John Chapin. Adolescents and Cyber Bullying: The Precaution Adoption Process Model. Education and Information Technologies,2016,21(4).、
- Bonanno Rina A,Hymel Shelley. Cyber bullying and internalizing difficulties: above and beyond the impact of traditional forms of bullying.. Journal of youth and adolescence,2013,42(5):685-697
- please highlight the knowledge gap.
Author response: Thank you for your suggestions. According to your suggestions, we have added the knowledge gap in the introduction.
As a new form of aggression, cyber aggression is more likely to occur and more harmful than traditional forms of bullying because of its universality and concealment. cyber aggression can cause physical and mental harm to the victims, such as depression, anxiety and even suicide (Bonanno & Hymel, 2013).It is reported that 10 to 21.9 percent of college students have committed cyber aggression, and 55.3 percent have been attacked (Gahagan & Vaterlaus, 2016), which seriously endangers college students' study and life. Therefore, it is necessary to further explore the mechanism of cyber aggression, so as to provide some useful suggestions for reducing college students' network cyber aggression and creating a good network environment.
Through literature review, we find that there are few studies on the influence of Just-world beliefs on aggressive behavior in China, especially the research on the mechanism of the influence of Just-world beliefs on aggressive behavior. It is also unknown whether the results obtained by foreign scholars are applicable in our country. Therefore, this study takes Chinese college students as subjects to explore the relationship between Just-world beliefs and aggressive behavior in the Chinese context.
- Can you shorten the introduction?
Author response: Thank you for your suggestions. According to your suggestions, We have cut down the repeated arguments in the introduction and the relevant content has been readjusted.
Introduction
The creation of the Internet is of major importance in the development of human civilization. The term “cyber aggression” refers to hostile behavior caused by inhibitive characteristics, and involves the use of networks or mobile electronic devices to write obscene or insulting communications that provoke, threaten, and harm a person or group [1]. Unlike the direct use of objects to cause physical injury, cyber aggression involves threatening and abusing others through obscure texts or negative pictures/symbols; this leads to psychological pain and long-term negative effects [2]. College students are common users of network platforms, and cyber aggression occurs frequently in college students. It is found that the incidence of network attacks among college students is about 60% [3], and about 55% of college students have suffered network attacks[4]. Cyber aggression can cause many physiological and psychological problems, such as anger, depression, anxiety, eating disorders, alcohol addiction, and even suicidal tendency [5] [6] [7]. However, compared to traditional forms of aggression, cyber aggression has been least studied[8]. Therefore, it is necessary to further explore the psychological mechanisms underlying cyber aggression. As well as helping to provide a psychological foundation to guide network supervision and network behavior, this would help to improve network-related moral reasoning in college students, improve the moral environment of network use, and develop a better online public mental health environment.
Individuals who believe in a just world firmly believe that the world they live in is just, and that their own and others’ gains and losses are determined and distributed by the world. Belief in a just world can provide individuals with a sense of security and control, and can moderate individual cognition, behavior, and emotion [9]. Generally, individuals with high levels of belief in justice are more inclined to believe that the world they live in is just, always have high expectations of life, are able to maintain a positive attitude about the future, voluntarily struggle for an ideal life, complete goals in a way that conforms to social norms, and strive to avoid all problematic behaviors that hinder the realization of their goals [6][10][11][12].However, real life is often characterized by injustice. Serious threats to an individual’s belief in self-justice reduce their sense of control over life and the surrounding environment and cause anxiety, anger, and hostility, thus stimulating aggression [13] [14] [15]. Moreover, individuals with low levels of belief in justice may use irrational cognitive strategies to reconstruct a sense of justice at the cognitive level. That is, such individuals tend to re-evaluate and interpret a victim’s character, morality, and other aspects, to try to determine the cause of their unjust treatment. During reconstruction of justices, a hostile attitude toward others and attribution bias activate individuals’ implicit aggression schemas. In the network environment, this implicit aggression schema becomes more and more intense because of the anonymity and freedom of network communication, which induces more cyber aggression [16] [17].
Self-control is the ability to restrain impulsive thoughts, regulate negative emotions, and regulate personal behaviors [18]. Research indicates that people with low self-control are more likely to show problem behaviors, such as crime, drug abuse, and alcoholism; in contrast, people with high self-control have healthy eating habits, good interpersonal relationships, and fewer diseases [19] [20] [21] [22]. Denson [23] et al. identified self-control failure as an important predictor of aggression, experiments have shown that in a provocative situation (noise interference caused by others), individuals who receive self-regulation training feel less angry and retaliate against others less. The dual-process model of self-control posits that when faced with temptation, individuals are prompted to make an emotional response to satisfy their impulsive desires; at the same time, they are stimulated to consciously evaluate and control the consequences of their impulses, however, if the intensity of the impulse motivation exceeds the cognitive effort to resist or regulate the impulse, the individual’s attempts at self-control fail; this causes a series of problem behaviors [24]. For example, an individual with a fluctuating or unstable belief in justice will often use the cognitive model of “everything that happens is reasonable” to rationalize the unjust events that happen to victims. In addition, they will often use slander, ridicule, satire, and other methods to argue that others deserve their punishment, to maintain long-term stability of their belief in justice. Over time, this type of hostility to others uses more cognitive resources. When it exceeds the limit of self-regulation, it makes individuals less sympathetic. Moreover, in the network environment of free speech and anonymity, individuals may express their impulses as cyber aggression to vent their emotions [3] [16] [25].
Some studies have also identified differences in cognitively ambiguous cues between male students and female students. Compared with women, men who are treated unjustly are more motivated to seek revenge when using the Internet and show a more obvious tendency to attack [26]. This could be explained by evolutionary differences. Generally, men are expected to be more competitive and in control, whereas women are expected to be more obedient to norms, honest, and kind. In interpersonal communication, women are more likely to care for and understand others, are willing to cooperate with others, and show more altruistic will; in contrast, men are more concerned about personal interests, tend to compete with others, and are more likely to use irrational cognitive strategies to maintain their belief in justice when experiencing cognitive distress over the treatment of others. Women often use self-regulation to control the development of risk-taking behavior, and transform social control into self-control, thus reducing the likelihood of crime; thus, men show more aggression [27].
To summarize, the belief in a just world theory, the dual-process model of self-control, and the GAM were used to generate the following three hypotheses, which were tested in this study. H1: The level of college student cyber aggression will be positively affected by a high level of belief in justice. H2: There will be an obvious indirect effect (mediating effect) of self-control; H3: The mediating effect of self-control will be moderated by gender. A hypothetical model was generated and is shown in Figure 1.
Methods
- Why did you select this type of sampling?
Author response: Thank you for your suggestions. The reason why I choose college students as the samples in this study is that college students are the common users of network platforms. And it is reported that 10 to 21.9 percent of college students have committed cyber aggression, and 55.3 percent have been attacked (Gahagan & Vaterlaus, 2016), which seriously endangers college students' study and life. Therefore, it is necessary to further explore the mechanism of cyber aggression, so as to provide some useful suggestions for reducing college students' network cyber aggression and creating a good network environment.
- How did you calculate the sample size?
Author response: Thank you for your suggestions. We did not estimate the size of the sample before collecting the questionnaire, which is an oversight of our research. But the sample size we collected was large enough as the sample size collected was more than 10 times of the item, which could achieve a good effect size.
- Where are the inclusion and exclusion criteria?
Author response: Thank you for your suggestions. The inclusion criteria in the study:the sample of the study should be college students in grades 1-4. And the exclusion criteria are that the sample of the study should not have substance abuse and similar experience of psychological investigation.
- elaborate more on the measurement tools. What is the total score of each tool and what does it indicate?
Author response: Thank you for your suggestions. We have added more details of each tool.
4.1 The Just-world belief scale
The Just-world belief Scale developed by Dalbert (1999) was used. The scale consists of two subscales: General Just-world belief Scale (GBJW) and Personal Belief in a just world Scale (SBJW). General Just-world belief Scale measures people's judgments about the fairness of what happens to others, and the Personal Belief in a just world Scale measures people's judgments about the fairness of what happens to them.
The scale comprises a total of 13 items. The Personal Belief in a just world Scale belief in self-justice subscale consists of seven questions; for example, “I believe most things that happen in my life are fair.” The General Just-world belief Scale comprises six questions; for example, “Generally others are treated fairly.” Responses are scored on a 6-point Likert scale,1 indicates complete disagreement, and 6 indicates complete agreement, on a scale from 1 to 6; The average score of each item of the scale was obtained,higher scores indicate higher levels of belief in justice. The α coefficient of the scale is 0.90, and the α coefficients of the two subscales are 0.84 and 0.87, respectively.
4.2 Self-Control Scale
The SCS-19 College Student Self-Control Scale developed by Tangney et al. (2004) and revised by Guo Yongyu et al. was used. The scale comprises 19 questions including five dimensions include resisting temptation, healthy habits, limiting entertainment, impulse control and focus on work,of which 4 (questions 1, 5, 11 and 14) are forward-scoring questions, while the remaining 15 items were scored in the reverse direction. Responses are scored on a 5-point Likert scale, the item options were "completely inconsistent, somewhat inconsistent, uncertain, fairly consistent, and completely consistent," rated 1, 2, 3, 4, and 5, respectively; higher scores indicate greater self-control. The α coefficient of the scale in this study was 0.85, and the McDonald's omega coefficients is 0.84.
4.3 Cyber aggression scale
The Scale for Adolescent Internet Deviance (sAID) compiled in Li Dongmei's (2008) doctoral dissertation was adopted , the scale contains three basic dimensions, a total of 35 items: 20 items of online aggressive behavior, 9 items of online pornographic behavior, and 5 items of online cheating. Among the three dimensions, Internet aggressive behavior is subdivided into four dimensions, including 6 items of aggression, 5 items of irritability, 5 items of hostility, 4 items of conflict. And in this study, only one dimension, namely aggression including 6 questions in total, is selected, for example, “On the Internet, I will say something on purpose to make others sad.” Responses are scored on a 5-point Likert scale, ratings range from 1(never) to 5(always); higher scores indicate more frequent cyber aggression. The α coefficient of the scale in this study was 0.87.
- the procedure of data collection needs more details. e.g., how did you approach the target population and recruit your sample?
Author response: Thank you for your suggestions.Random sampling method was adopted to test college students (grades 1-4) in four colleges and universities in Anhui and Henan. A total of 1308 questionnaires were distributed on site by class. After removing invalid questionnaires such as not recovered, lost in the middle, missing, misfilling and missing values, 1136 questionnaires were sorted out.
- Where are the ethical considerations of your study? e.g., IRB approval and informed consent procedures?
Author response: Thank you for your suggestions.There is an informed consent process in our research.
discussion
- State the clinical implications and recommendations for future research
Author response: Thank you for your suggestions. With regard to clinical implications, Cyber aggression response is a systematic project, which requires the joint efforts and all-round cooperation of the government, schools and families .The government should establish and improve the network laws and regulations to combat cyber aggression; schools should do a good job in technology monitoring, filtering aggression information, and provide timely psychological counseling for students who experience cyber aggression; parents should manage their minor children's mobile phone use and educate them to use the Internet properly.
In view of the shortcomings of this study, later research can expand the region of research object selection and increase the representativeness of research samples. In order to ensure the objectivity and accuracy of the research results, the follow-up research can consider combining behavioral experiments and ERP experiments. In the subsequent research, intervention research can be adopted to provide a reliable theoretical guarantee for educational practice.
- highlight the strengths of your study before the limitations and think of more limitations
Author response: Thank you for your suggestions. According to your suggestions, firstly, we have pointed out the educational significance of this study before reflecting on the shortcomings, which are the strengths of our study. Secondly, we have added some more limitations:
- Due to the limitations of practical conditions, the objects of this study only involve college students from four universities in Anhui Province, so the samples can not reflect the overall characteristics well.
- This study only discusses the influence of gender and self-control on the relationship between belief in a just world and cyber aggression behavior, but it is only a preliminary exploration. Many questions still need to be solved, such as the mechanism of self-control and further intervention research.
- Participation in cyber aggression may have been underreported because of the tendency of individuals to provide socially desirable answers. In addition, recall bias may have happened, which may also have caused a certain bias in the study results.
conclusion
1.revise your conclusion to be stated in a narrative way
Author response: Thank you for your suggestions. According to your suggestions, we have revised our conclusion and transformed it into a narrative way.
The revised version:
After controlling for additional variables such as age and grade, this study found that belief in a just world can significantly negatively predict college students' cyber aggression behavior, and self-control plays a mediating role between belief in a just world and cyber aggression. And The direct predictive effect of belief in a just world on cyber aggression and the mediating effect of self-control on the relationship between belief in a just world and cyber aggression are both moderated by gender.
Round 2
Reviewer 1 Report
Dear authors: Thank you very much for the changes made as a result of the indications made in the first review. However, the McDonald Omega coefficient would still need to be included in the other two scales used in the study. I also suggest including other indices or coefficients such as composite reliability and the average variance extracted in each of the scales or instruments used.
Author Response
Thank you for your suggestions. Based on your comments, we added the McDonald Omega coefficient of the two scales used in the study. Please see the Research tool section of our revised manuscript.
Reviewer 2 Report
The authors have addressed all of my comments. Good Job
Author Response
Thank you for your comments and suggestions.